# Morphostructural Differences between the Historical Genetic Lines of the Spanish Merino Sheep

**DOI:** 10.3390/ani13020313

**Published:** 2023-01-16

**Authors:** Antonio Granero, Gabriel Anaya, María J. Alcalde

**Affiliations:** 1Asociación Nacional de Criadores de Ganado Merino (ACME), 28028 Madrid, Spain; 2Department of Genetics, University of Córdoba, CN IV, KM 396, 17071 Córdoba, Spain; 3Agronomy Department, University of Sevilla, 41013 Sevilla, Spain

**Keywords:** Merino, zoometry, ancestral, genetic lines, diversity

## Abstract

**Simple Summary:**

This study addresses the analysis of the morphometric differences based on quantitative traits in the different historical genetic lines of the Spanish Merino sheep breed. The results show the existence of significative differences both in the measurements taken and in the indexes calculated between the different genetic lines. This result could indicate how different genetic lines within the autochthonous Merino breed have developed due to the chosen management models and the adaptation to the specific ecosystems in which the sheep have been bred in pure, closed conditions for centuries, which have been maintained to the present day. Our study complements a previous genomic approach, in which the different historical lines of the breed were characterized, showing the genuine genetic pool of the Spanish Merino, the origin of all the Merino and Merino derived breeds in the world.

**Abstract:**

The Merino breed, which originates from Spain, is the most emblematic livestock breed in the world, since it is the first with a worldwide extension and has had an important impact on the genetic origin of several of the main current sheep populations. For this reason, it is of vital importance to typify the historical genetic lines of the original Spanish Merino breed and thereby ensure the conservation of its variability. In the present study, we used 337 purebred animals (males and females) registered in the Genealogical Book of the Native Merino Breed. All the animals were descendants of herds from six ancestral genetic lines (Maesso, Egea, Granda, López-Montenegro, Hidalgo, and Donoso). Significant differences were found in all the morphometric traits and indexes between the different genetic lines. Using discriminant analysis, 84% of the animals were classified correctly into their historical genetic lines. Furthermore, the distances between the lines, calculated by a cluster test, showed that Hidalgo, Maesso, and Donoso had the most clearly defined lines, while the Granda, López-Montenegro, and Egea lines were more similar to each other. All this demonstrates the rich genetic variability existing in the genuine gene pool of the Merino sheep breed.

## 1. Introduction

The Spanish Merino sheep is considered the most important breed worldwide not only for its history, census, and expansion, but also for its role in the influence of the appearance of new and/or derived breeds [1]. However, the origin of the Merino breed is a hotly debated topic, for which several theories have been proposed [2,3]. According to the most widely accepted theory, the breed was established in the Iberian Peninsula by the Romans through the crossbreeding of three breeds of sheep from North Africa, Italy (Tarentum), and the Iberian Peninsula [4]. The resulting animals had fine wool, and they spread over centuries throughout the Iberian Peninsula and other parts of the world, such as Southern Italy (15th century) or Germany and France (18th century), where the Merinolandschaf and Rambouillet breeds were founded [5]. In addition, other breeds were derived from Merino stocks and developed as mutton breeds, such as the French Ile de France and Berrichon du Cher [5]. The dispersal of Merino sheep in Eastern Europe also started in the 18th century, and during the 19th century, the Merino breed was also exported to Australia and New Zealand. In more modern times, Merino and their derived breeds have become cosmopolitan [4,6].

In Spain, the economic crisis of the 1960s saw a shift in the selection criterion towards meat production instead of wool production. Some breeders began to use foreign breeds with higher meat aptitude to cross with native Merino mother populations which were not being bred in purity [7]. As a result, several ancient Spanish genetic lines disappeared as these herds were turned exclusively to the production of commercial meat [8]. However, some of the historical lines have survived thanks to the efforts of several groups of traditional breeders who decided to continue selecting individuals for their wool quality, rusticity, and fertility in closed herds, just as they had been bred over the previous 200 years. Among them, the most important pure Merino lines selected for wool production, rusticity, and fertility that are still bred in Spain are the Maesso, Egea, Granda, López-Montenegro, Hidalgo, and Donoso [9,10,11,12]. These lines are the only remaining ancestral sheep from the original Merino and are currently bred in just 11 herds. However, these populations are not officially acknowledged as strains. The zoometric study of their local resources (characterized by their adaptation to adverse edaphoclimatic conditions, resistance to disease, and consumption of poor quality pastures, among others) is necessary, since it provides useful information about racial characterization, allowing us to measure the productive capacities of individuals or the inclination towards a certain zootechnical method of production, as well as detecting genetic relationships between breeds in different domestic species [13]. Their corporal composition has been influenced by the environment and management methods, so body measurements are a reliable way to detect selection signatures between the lines.

In the conservation of genetic resources, the main objective is to preserve variability within populations, following the hypothetical correlation between genetic variation and population viability [14]. Moreover, several years of selective breeding of different populations has led to phenotypic changes and genetic adaptation to various environmental conditions [15]. Studies are therefore necessary to characterize and differentiate populations, and the origin and history of the groups should be documented (including the genetic line and linages) [16].

The morphological characteristics of the different animals are qualitative, while the structural characteristics are quantitative and, therefore, possible to measure, with morphostructural variables which can be treated statistically [17]. These six historic genetic lines have been previously characterized with a genomic approach [12], showing considerable genetic differences between them, even though they belong to the same breed. The genomic results also revealed a different selection footprint compared with the modern Merino, together with the existence of more ancestral selection processes in these genetic lines compared to modern Merino sheep, demonstrating that many generations were selected earlier to avoid the current practice of crossbreeding [12]. The present study therefore aims to evaluate the morphological characteristics of these Spanish Merino lines, in order to complete the genomic approach and characterize the lines, showing the large morphological differences found within the Merino racial pattern. In this way, the foundations can be laid for them to be officially recognized as genetic lineages.

## 2. Materials and Methods

### 2.1. Animals and Measurements

For the study, we used 337 purebred animals (males and females, approximately 1 year old) registered in the Genealogical Book of the Native Merino Breed. All the animals were bred in similar conditions on 11 farms that belonged to the historical genetic lines of Donoso (31), Egea (40), Granda (60), Hidalgo (80), Maesso (60), and López-Montenegro (66).

A total of 22 measurements were taken for each of the animals in spring, just after the shearing process. The measurements are described in Table 1, Table 2 and Table 3.

The Body Weight (BW) was measured with a dynamometer, and the animals’ weights in kg were recorded.

Measurements were taken on the animals’ left side, with the four feet firmly planted on the floor on a horizontal plane. After that, seven indexes were calculated using some of the measurements to obtain the relationships between 2 lineal dimensions [18,19,20]:

Body Index (BI): (LD/ThP) × 100.

Compactness Index (CoI): (BW/WH) × 100.

Pelvic Index (PeI): (RW/RL) × 100.

Thoracic Index (ThI): (BD/DSD) × 100.

Cephalic Index (CeI): (HW/HL) × 100.

Proportionality Index (PI): (WH/LD) × 100.

Load Cannon Bone Index (LCI): (ACP/BW) × 100.

### 2.2. Statistical Analysis

For the statistical analysis, we used the IBM SPSS Statistics 25.0 software for Windows (IBM Corp., Armonk, NY, USA). An analysis of variance (ANOVA) was conducted using the gender and historical genetic lines (Donoso, Egea, Granda, Hidalgo, Maesso, López-Montenegro) as fixed effects. Significant differences between the average values were determined by a post hoc Tukey test. Next, we calculated a principal components analysis (PCA) to define the underlying structure among the variables in the analysis, saving the factorial scores as variables and creating a new variable for each factor in the final solution using the Regression method. After that, a discriminant analysis (DA) was carried out using a stepwise model considering the genetic lines of the animals as a classification factor, introducing in each step the variable that minimizes the global Wilks lambda. The aim of this step was to estimate the proportion of animals that were properly classified into their line, using the discriminant classification method of leave-one-out cross-validation. Finally, we generated a cluster analysis (a simple linkage based on Euclidean distances) and plotted it as a dendrogram.

## 3. Results

The results of the measurements are shown in Table 4. In general terms, significant differences were found between the different genetic lines. In all the measurements, the results were significantly different in the gender factor, showing clear sexual dimorphism in these lines, with important differences between males and females. There are also major differences depending on the lines analyzed. Despite the sizeable differences based on gender, the results indicated that both males and females had uniformity in each line, differing from the other lines in the same way, due mainly to the fact that, in the different sections of this work, all the animals were considered in the other analyses.

### 3.1. Head

Concerning the parameters measured in the head, the lowest values for the HL, HW, WF, and WCh measurements corresponded to the Maesso line and the highest values, except WCh, were shown by the Hidalgo. However, the WF measurement is of particular interest because its point of reference is its distal part, and in the case of the animals of the Hidalgo line, the end of the tuft did not usually end in a straight line but in a kind of peak, in the form of wool-free patches on either side of the head. As regards the other lines, López-Montenegro obtained the second lowest value for HL and WF and the third lowest for HW, while Donoso, Egea, and Granda, with values close to the extremes depending on the measurement, were in the intermediate zone. As for EL, Donoso, followed by Maesso, obtained the lowest values and Hidalgo the highest values, while the values obtained by López-Montenegro, Granda, and Egea were intermediate.

### 3.2. Trunk and Extremities

As for the trunk and limbs, the lowest values of the measurements SW, NL, WH, BD, HSF, DSD, LD, LBL, RH, RW, RL, BuW, BL, ThP, ACP, PCP, and BW were found in the Maesso line, followed by López-Montenegro, except in the values HL, WH, HSF, and LBL. In contrast, the highest values of SW, NL, WH, BD, HSF, DSD, LD, LBL, HR, RL, BL, and BW were found in the Hidalgo line and the highest values of RW, BuW, ThP, ACP, and PCP were in the Donoso genetic line. Egea and Granda showed intermediate values.

### 3.3. Zoometric Indexes

The indexes derived from the measurements taken are shown in Table 5. All the indexes showed significant differences in all the lines and gender. The Donoso line showed the lowest values in the Body and Thoracic indexes and the highest in the Pelvic and Proportionality indexes. In contrast, the Hidalgo line showed the highest values in Body, Compactness and Thoracic indexes and the lowest in the Pelvic, Proportionality and Load cannon bone indexes. According to the Cephalic index, the animals were classified into two clearly defined groups. The lowest values appeared in the Donoso and Hidalgo lines, while the highest values were found in the Egea, Granda, Maesso, and López-Montenegro. The highest values of the Load cannon bone index were seen in the Maesso line.

### 3.4. Principal Component Analysis (ACP)

Table 6 shows the Principal Component Analysis (PCA) based on the average values of all the morphometric measurements for the six genetic groups. The coefficients show the relative contribution of each measurement to a specific principal component (factor), while the percentage of the total variance was used to determine how the total component solutions account for the variables (measurements) represented. The analyses showed that the first two factors contributed to 80.81% of the total variance, and that the first principal component accounted for 73.80% of the total variance. All the parameters made a very high positive contribution to the variation and differentiation of the genetic lines. The second factor accounted for 7.01% of the total variance, where the principal contribution was made by the Ear length.

Figure 1 shows the graphic representation of males and females analyzed in the two-dimensional space generated by the two main components, according to their factorial scores using the Regression method. Groupings in the different lines can be seen, which are more clearly differentiated in females than in males. The Donoso line was located separately from the other genetic lines.

### 3.5. Discriminant Analysis

A discriminant analysis (DA) was carried out, using a stepwise model considering the animals’ historical genetic lines as classification variables. The percentages of correct assignments can be observed in Table 7 for the different cases.

All the measurements and indexes were entered as variables, with 17 variables used in the statistical analysis in the following order: EL (Ear length), BuW (Buttock width), HSF (Height at substernal foramen), HW (Head width), CoI (Compact index), WCh (Wool extension on cheeks), ThP (Thoracic Perimeter), BD (Bicostal diameter), NL (Neck length), HL (Head length), BL (Buttock width), BW (Body weight), WH (Withers height), LCI (Load cannon bone index), RH (Rump height), BI (Body index), PI (Proportionality index). 84.0% of the animals were classified correctly. All the animals from the Donoso line were correctly classified. The group of Maesso and Hidalgo animals were classified with a high degree of accuracy of 96.7% and 95.0%, respectively. In the case of the López-Montenegro and Egea animals, the values were a little lower (80.3% and 70.7%, respectively) while the animals from Granda line were those with the lowest value of correct classification (61.7%).

### 3.6. Cluster Analysis

The dendrogram showed how Egea and Granda were the closest lines according to the morphometric characterization, while Maesso appeared as the family most distant to the other genetic lines as regards corporal conformation (Figure 2). Intermediate levels of proximity appeared in the Donoso, López-Montenegro, and Hidalgo lines, with Hidalgo being the farthest within the intermediate group.

## 4. Discussion

The different historical genetic lines of the Spanish Marino have previously been studied using a genomic approach, in which high levels of genetic differences were found between the populations analyzed, as well as high variability within the breed [12]. However, although there are evident phenotypical patterns between these ancient groups that has been previously described [21], this is, to our knowledge, the first morphometric study carried out in the historical genetic lines of the Spanish Merino.

Therefore, the data obtained could not be compared with any previously described measurements, except for the WH, 72.5 cm, and the BW, 90 kg, (mean for both sexes) published in the National Breed Information System (ARCA) of the Ministry of Agriculture [22], or with the WF 58 cm, RH 53.46 cm, HSF 32.46 cm, LD 60.93, and ThP 80.97 obtained in the biometric study carried out by Diaz Montilla [23] on a group of one-year-old Merino sheep.

As for the ARCA data, the mean of all the animals studied of the six historical lines showed a WH value of 69.92 cm, and a BW of 59 kg, calculated for both sexes. In the case of the WH, the measurement was similar, with a small difference of only 2.58 cm. However, the opposite occurred in the BW, where a significant difference of 31 kg between the data was found. In relation to the results obtained by Diaz Montilla [23], these differ significantly from the means obtained in the present work for WF 69.92 cm, RH 71 cm, HSF 40.22 cm, LD 71.28 cm, and ThP 90.09 cm. In this context, when comparing the measurements, we obtained the following differences: WF 11.92 cm, RH 17.54 cm, HSF 7.76 cm, LD 10.35 cm, and ThP 9.12 cm.

Regarding the differences with the ARCA data, this could be due to the impact made by the Hidalgo lineage, which was commonly used to “re-merinize” crossed Merino flocks. When hybrid Merino flocks are crossed with pure Merino rams, the resulting F1 shows a high manifestation of pure Merino traits (e.g., fine, dark-colored wool due to its high fat content). This fact has had a great impact on the current population of the breed in Spain; therefore, the data published in ARCA, although quite high for the breed according to the results obtained in this study, is closer to the average for the Hidalgo BW, 73.35 ± 1.23 kg, than to the average for the six historical lines. In fact, one of the purposes of the present work is to update and complement the ARCA data, in order to produce an integrated, scientifically contrasted zoometric characterization for the breed.

Regarding the differences with the measurements obtained by Díaz Montilla [23], these could be due to the impact of changes in the exploitation system of the Merino breed in Spain. Thus, according to Esteban [24], the Merino breed has traditionally lived in very harsh environments under a highly extensive exploitation regime, and for over 75 years (the time difference between the two studies), the breed has been subjected to a better diet and favorable management, which has led to the increase in size reflected in the present study. However, we cannot ignore the fact that Díaz Montilla’s [23] study only included females, which could also explain the differences with our data, which were obtained from both sexes. On the other hand, when we compare the measurements taken by Díaz Montilla [23] with those obtained in this work for the Maesso line, we can see that WF 61.43 cm, RH 62.38 cm, HSF 34.36 cm, LD 62.98 cm, and ThP 79.07 cm tend to converge, which significantly reduces the differences in the comparison with the joint means of the six lines WF 3.43 cm, RH 8.92 cm, HSF 1.9 cm, LD 2.05 cm, and ThP −1.9 cm. Finally, although they are similar measurements, it remains unclear whether Díaz Montilla [23] used the same methodology to obtain his data as that used in our study.

The results of the measurements showed clear differences between the different lines. Here, Maesso was the genetic line with the smallest individuals of all, and the animals were classified as ellipsometric according to the Baronian scale [25]. It was followed by López-Montenegro, which could be classified as a eumetric line, but with a slight inclination towards ellipsometry, showing the second lowest values for almost all measurements including BW. The Granda, Egea, and Donoso lines showed similar intermediate values between them, with the order alternating according to the measurement, and they can clearly be characterized as eumetric animals. Finally, Hidalgo showed the highest values for almost all the measurements including BW, especially those related to body proportions; based on the average weight obtained, animals from this line could be characterized as hypermetric within the Merino breed.

The weight of autochthonous breeds is conditioned by the environment and management in which the animals live, which can give rise to different ecotypes [24]. The morphological differences found in the different genetic lines analyzed in this study suggest that the selection carried out in these herds over the centuries, both natural and owner-directed, has led to this rich variability of each group of animals, with different characteristics evident within the same breed, always following the Merino breed pattern. In this context, over the centuries, the different breeders of each of these lines have endeavored to breed a type of animal with specific morphological characteristics and related productive and behavioral aspects which meet their needs, while always seeking full adaptation to the environmental conditions in which they live, without ignoring the genetic derivative that they may have due to reproductive isolation [12]. In fact, the greater differences found between the Maesso and Hidalgo lines could be due to their differing degrees of adaptation to specific productive systems [26].

In this context, the usual practice on Maesso farms has always been to ensure that the replacement animals are small and hardy and can move about efficiently despite the poor resources provided by the southern mountainous areas where their herds live. In particular, it is essential that they are well adapted to the high summer temperatures, following Bergmann’s rule [27], by which being smaller allows animals to regulate their internal temperature better and in general be more resilient [28]. In this way, in each generation, animals born with a smaller size will be better adapted to their environment than the others. In our study, the Maesso animals showed the lowest data for all measurements except for EL (see Table 4). The classic management technique used with these herds for decades has been to divide the animals into two groups, which are kept apart from each other. One group is made up exclusively of pure, registered Maesso animals, both male and female, and replacement, renewal, or re-breeding sheep are selected only from this group. After their first calving, the pure-bred females of this group are transferred to the herds that make up the other group and are crossed with Merino Precoz animals to generate a hybrid, unregistered animal which is destined solely for meat production. The objective is to maintain the hardiness and adaptability to the environment of the pure line with the females, and to be used as a gene pool to generate crossed, unregistered animals for commercial purposes, due to the high meat efficiency supported by the genetic properties of the Merino Precoz males.

The Hidalgo line is made up of animals originally from León (Northern Spain) with a transhumant character [29]. However, in recent times, the migratory habits of some herds have been replaced by permanent management on farms. In León, due to the low cost of feed during this period, it has become fashionable to select larger animals, ignoring, on the other hand, relevant aspects of conformation for meat production, with the aim of producing larger offspring which can quickly reach slaughter weight as suckling lambs. In the case of the Hidalgo herd, the sheep experienced extremely favorable breeding conditions, as they enjoyed the tranquility of the Extremadura pastures during the autumn, winter, and spring seasons, while in summer they grazed in the best mountain areas of León, where they enjoyed abundant pasture and a less extreme climate. All of this contributed to the size gain of the Hidalgo sheep compared with the other lineages (see Table 4). For this reason, the Hidalgo strain showed the highest values related to the body size. However, in the measurements more related with the carcass conformation to meat aptitude (RW, BuW, ThP, ACP, and PCP), the values of the Hidalgo line were lower than those showed by the Donoso line, probably because it is a commoner practice in the region of León (with a high census of Hidalgo animals) to slaughter the lambs during lactation, while in Extremadura, where there is the highest representation of Donoso line, the animals are usually slaughtered when they are older.

As regards the López-Montenegro, Granda, Egea, and Donoso lines, although the data show a greater homogeneity, certain factors must be taken into account. The slightly lower values of the López-Montenegro line are largely due to the natural selection system that has historically been applied by breeders to obtain animals which are as well adapted as possible to the environment in which they live. In this way, this line has always avoided making a strong selection for meat traits in order not to lose its genetic heritage. On the other hand, in the Granda, Egea, and Donoso lines, it can be seen from the data that a selection has been made to adapt a rustic breed to market demands in terms of the conformation of animals for meat production, but without losing sight of their historical origin [30]. It is therefore worth highlighting Donoso as the most suitable from the point of view of meat conformation, showing more cylindrical conformations with the highest value for ThP and the highest values for ACP and PCP. Given that the cannon bone circumference is a measurement related to the animal’s silhouette [19], it has a differentiating value between breeds destined for dairy and meat production. For instance, the former tends to have medium-thick or thin shanks, while the latter tend to have medium to large ones [17].

The indices were calculated with the purpose of determining somatic states predisposed to certain functionalities. In this way, it was possible to show the relationships between certain elements of height, compactness and length generally used in zootechnics to estimate the animals’ proportions and conformation. The characters obtained from head measurements, such as the cephalic index, are of great ethnological importance, because their variation is not influenced by environmental factors or by the handling of the animals [17]. The animals in this study could be classified as dolichocephalic, although the animals of the Donoso and Hidalgo lines had a lower index than the other lines. Regarding the body index, which according to [31] gives an estimation of the proportionality of the breed, the animals of all the lines presented BI values are equal to or higher than 85, characterizing the breed as brevilinear or compact. However, different levels within this classification were also observed. Except for the Donoso line, the rest presented a higher format than the Australian Merino [32]. Regarding the thoracic index, there are variations in the shape of the thoracic section, being higher (more circular) in meat sheep and lower (more elliptical) in dairy sheep. For the sample studied, a different significance was obtained within the brevilinear classification (ThI ≥ 89). However, without being specifically selected for this, the Hidalgo line showed a more elliptical thoracic section, which could be related to the larger size of its udders compared to other historical strains. In this context, in a breed with a dual aptitude for meat and wool, the Hidalgo line have a greater predisposition for milk production within the limited possibilities of the breed. The animals with the highest Compactness index are those of the Hidalgo line, which are the heaviest animals in relation to their height. The Pelvic index indicates the relationship between width and length of the rump, which reflects in all the genetic lines a rump which is proportionally wider than long and a convex pelvis, which is associated with reproductive functionality (easy lambing), in contrast to the Australian Merino [32]. The Load cannon bone index shows the harmony between the total body mass of the animal and the conformation of the limbs, i.e., the greater the weight, the greater the degree of robustness. Therefore, the Maesso line has the greatest strength.

Principal component analysis (PCA) is used as an interdependent technique to identify morphometric parameters that best serve as breed-specific markers [34]. The PCA assay has been used to extract factors from the body measurements that contribute to the morphometric variations among individuals [33,34,35,36]. The two principal components account for 80.81% of the variations in the evaluated traits. Salako [37] used PCA for 10 linear body measurements and reduced them to two principal components that accounted for 75% of the total variation, while Marković et al. [35] used PCA for 10 measurements and 10 indexes, with three principal components accounting for 96% of the total variation. In both cases, the studies focused on differentiation between breeds, whereas in our study since the focus was on genetic lines, the variables studied provided us with a good differentiation between lines. In our case, in the first principal component (73.80%), all the parameters showed a very high positive contribution to variation and lines differentiation, which suggests they are correlated. Since PCA was used as a method for reducing the number of variables, only the WCh value (lower than 0.7 for both factors) did not provide enough information to differentiate between the lines in our study. The morphometric traits in the same component were classified together, and so we can conclude that they probably have common genomic positions for their genetic control [33]. In our study, the proportion of variance accounted for in the original variables was very high, from 0.720 to 0.965. According to the final communality, the variables explained by the set of factors retained by the model are well explained when they are greater than 90%.

The accuracy in the classification using discriminant analysis (DA) was very high. Interestingly, the variable with the highest discrimination impact was Ear length (matching the results of Yunusa et al. [36]), while the index which best discriminated between the genetic lines was the Compactness index. Animals from the Donoso line were clearly different from the rest, when all the individuals were assigned to their specific population, and no animals from other lines were ascribed to the Donoso line. Based on the variables studied, the differences between the lines are evident. This result reinforces the findings of Granero [12], in which the Donoso line was the most genetically differentiated compared with the others, according to the PCA and the genetic structure of the population.

In addition, the discriminant assay (Table 7) classified the Egea, Granda, and López-Montenegro historical lines with lower values compared to the others, perhaps due to the fact that all these lines are currently bred in the same geographical area and under similar management. It can therefore be assumed that these lineages could have been formed centuries ago from the same historical herds [12]. Additionally, the management used by the breeders of these lines has always aimed to conserve the variability within each of their closed herds, avoiding the high-pressure levels of selection so the animals could have more morphometric similarities within the herd although they are genetically different. In the Granda and Hidalgo lines, for instance, there is the particularity that on the same farm, a Granda and a Hidalgo herd have been raised separately for 25 years, from which we have taken the measurements for the present study. In this case, although the herds have been raised in a sealed way, the management of both may have been the same, and the same selection criteria may have been used in both herds.

The cluster analysis also establishes quite a few differences between the lines, with Hidalgo being the most different of all. In this case, the morphometric approach used in the results is different from the genomic approach. The genetic differences show the López-Montenegro and Granda lines are the most similar, and that the Donoso is the most differentiated line, rather than the Maesso line [12]. Generally, such phenotypic divergence among populations might be partly associated with differences in the production systems, agroclimatic conditions, and natural resources [38].

Morphometry has long been used as a classical approach to characterizing a breed based on its body conformation. However, it is currently in disuse, so there are no studies on the Merino or Merino-derived breeds that allow us to compare the phenotype of the pure Spanish Merino. We consider the results obtained in this work of great importance, since they compare the morphotype of the Spanish Merino with each of its pure genetic lines. Subsequent body measurements could serve as an indicator to ensure that the racial pattern of this ancestral breed, which forms the basis of its rich genetic reservoir, is maintained.

## 5. Conclusions

This study demonstrates the morphological variability that exists between the different historical genetic lines that make up the Spanish Merino breed. The confirmation of these phenotypic differences, added to the genetic differences previously found in these populations, allows us to define them as strains. It is clear that the high level of variability found in the breed is due to the presence of these historical genetic lines, which have remained pure thanks to the management that has been carried out in the same way for centuries. We can therefore affirm that the variability of the breed is guaranteed so long as the breeding of these historical lines continues, and its rich genetic reservoir is maintained.

## Figures and Tables

**Figure 1 animals-13-00313-f001:**
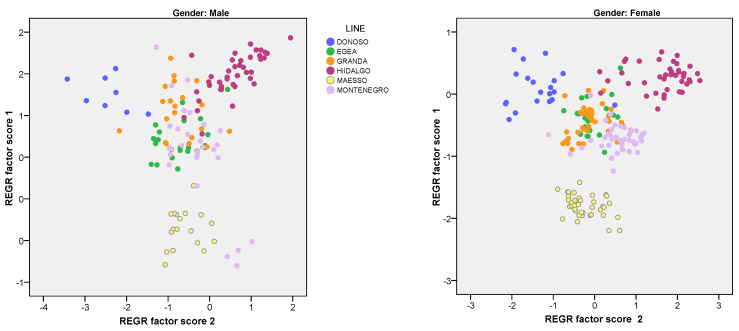
Scatter plot of the principal component analysis (Factor 1 vs. Factor 2), for females and males.

**Figure 2 animals-13-00313-f002:**
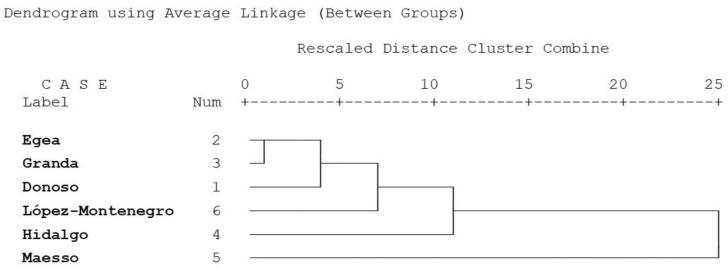
Dendrogram using Average Linkage between genetic lines.

**Table 1 animals-13-00313-t001:** Measurements taken with a measuring compass.

Head length (HL):	From the occipital protuberance or nape to the midpoint of the upper jaw.
Head width (HW):	Maximum distance between the two orbits.
Wool extension on forehead “bow” (WF):	Extension of wool covering the area from the occipital protuberance to the midpoint of an imaginary line passing below the eyes (frontonasal suture).
Wool extension on cheeks (WCh):	Extension of wool covering the cheeks, which goes from the lower point of insertion of the ear to the end where the wool ends.
Ear length (EL):	Rectilinear distance between the base of insertion of the ear and its free end.
Shoulder width (SW):	Maximum distance between the highest points of the scapulo-humeral joints.
Neck length (NL):	Distance between the lower point of insertion of the ear and the upper end of the point of shoulder.
Lumbar back line length (LBL):	Distance between the most declining point of the withers and the external iliac tuberosity (tip of hip).
Rump width (RW):	Maximum distance between the two external iliac tuberosities or tips of the hip.
Rump length (RL):	Distance between the external iliac tuberosity (tip of hip) and the ischial tuberosity (tip of the buttock).
Buttock width (BuW):	Maximum distance between the animal’s thighs.
Buttock length (BL):	Distance between the ischial tuberosity (tip of the buttock) and the upper end of the leg.

**Table 2 animals-13-00313-t002:** Measurements taken with the aid of a zoometric stick.

Withers height (WH):	Distance from the ground to the highest point of the withers (inter-scapular region).
Bicostal diameter (BD):	Maximum width of the thorax in a vertical plane that passes behind the elbow (5th rib).
Height to substernal foramen (HSF):	Distance from the ground to the substernal foramen.
Dorso-sternal diameter (DSD):	Distance between the most declining point of the withers and the sternal region behind the elbow.
Longitudinal diameter (LD):	Distance between the point of shoulder and the tip of the ischium.
Rump height (RH):	Distance from the ground to the highest point of the internal tuberosities of the ileum (apex of the first process of the sacrum).

**Table 3 animals-13-00313-t003:** Measurements taken using a flexible mete.

Thoracic perimeter (ThP):	Beginning at the most declining point of the withers, passing through the right side, sternum (immediately behind the elbow), left side, and returning to the withers.
Anterior and Posterior cannon bone perimeter (ACP and PCP):	Minimum perimeter of these cannon bones.

**Table 4 animals-13-00313-t004:** Means and standard errors for zoometric measurements taken for animals of the different historical genetic lines of Merino Sheep breed. Analysis of variance (*p*-value): gender and genetic line effects.

				Genetic lines			*p*-Value		
	Donoso	Egea	Granda	Hidalgo	Maesso	López- Montenegro	Line	Gender	Line × Gender
HL	27.15 ± 0.35 c	26.28 ± 0.31 b	26.01 ± 0.24 b	28.16 ± 0.18 d	24.48 ± 0.32 a	25.86 ± 0.27 b	<0.001	<0.001	<0.001
HW	11.66 ± 0.16 b	12.05 ± 0.10 d	11.94 ± 0.11 cd	12.29 ± 0.08 e	11.22 ± 0.11 a	11.80 ± 0.10 bc	<0.001	<0.001	<0.001
WF	15.52 ± 0.39 b	15.43 ± 0.19 b	15.52 ± 0.21 b	16.18 ± 0.22 c	14.05 ± 0.20 a	15.30 ± 0.22 b	<0.001	<0.001	0.004
WCh	8.40 ± 0.40 ab	9.68 ± 0.19 d	10.90 ± 0.18 e	8.76 ± 0.26 bc	7.69 ± 0.15 a	9.39 ± 0.21 cd	<0.001	<0.001	<0.001
EL	8.65 ± 0.15 a	10.52 ± 0.12 c	10.83 ± 0.09 c	13.21 ± 0.07 d	9.62 ± 0.08 b	10.86 ± 0.06 c	<0.001	>0.050	0.020
SW	21.97 ± 0.33 d	21.49 ± 0.34 d	20.62 ± 0.31 c	22.68 ± 0.29 e	17.18 ± 0.29 a	19.58 ± 0.31 b	<0.001	<0.001	0.044
NL	28.90 ± 0.08 b	30.00 ± 0.16 c	31.06 ± 0.25 d	34.64 ± 0.19 e	26.99 ± 0.24 a	30.61 ± 0.17 d	<0.001	<0.001	<0.001
WH	72.10 ± 0.62 c	70.90 ± 0.65 c	69.51 ± 0.53 b	74.78 ± 0.38 d	61.43 ± 0.64 a	70.82 ± 0.59 bc	<0.001	<0.001	<0.001
BD	32.50 ± 0.35 b	32.48 ± 0.30 b	32.18 ± 0.36 b	35.06 ± 0.27 c	29.15 ± 0.40 a	31.84 ± 0.34 b	<0.001	<0.001	<0.001
HSF	41.34 ± 0.47 b	40.73 ± 0.38 b	40.36 ± 0.26 b	43.30 ± 0.21 c	34.36 ± 0.32 a	41.24 ± 0.26 b	<0.001	<0.001	0.008
DSD	31.58 ± 0.33 d	31.30 ± 0.25 cd	30.63 ± 0.32 bc	32.99 ± 0.21 e	28.36 ± 0.36 a	30.10 ± 0.30 b	<0.001	<0.001	<0.001
LD	71.71 ± 0.76 c	71.94 ± 0.82 c	72.37 ± 0.72 c	78.92 ± 0.64 d	62.98 ± 0.55 a	69.81 ± 0.68 b	<0.001	<0.001	>0.050
LBL	30.52 ± 0.20 b	30.87 ± 0.37 b	30.20 ± 0.27 b	34.73 ± 0.71 c	26.63 ± 0.33 a	30.23 ± 0.21 b	<0.001	<0.001	>0.050
RH	71.85 ± 0.65 bc	72.64 ± 0.63 c	71.96 ± 0.60 bc	76.32 ± 0.34 d	62.38 ± 0.59 a	70.89 ± 0.52 b	<0.001	<0.001	0.001
RW	19.98 ±0.17 e	18.77 ± 0.15 d	18.26 ± 0.16 c	19.56 ± 0.08 e	16.22 ± 0.16 a	17.69 ± 0.16 b	<0.001	<0.001	<0.001
RL	24.13 ± 0.31 d	24.09 ± 0.26 d	23.42 ± 0.29 c	25.51 ± 0.20 e	20.47 ± 0.24 a	22.80 ± 0.24 b	<0.001	<0.001	0.007
BuW	28.81 ± 0.29 e	24.02 ± 0.29 c	24.04 ± 0.26 c	25.78 ± 0.18 d	21.06 ± 0.22 a	22.55 ± 0.28 b	<0.001	<0.001	>0.050
BL	18.47 ± 0.23 d	17.56 ± 0.22 c	17.45 ± 0.15 c	19.15 ± 0.18 e	13.73 ± 0.14 a	16.70 ± 0.24 b	<0.001	<0.001	0.002
ThP	99.42 ± 0.80 f	91.76 ± 0.62 d	89.42 ± 0.90 c	95.31 ± 0.46 e	79.07 ± 0.79 a	85.56 ± 0.80 b	<0.001	<0.001	<0.001
ACP	9.55 ± 0.13 d	9.09 ± 0.11 c	9.12 ± 0.09 c	9.41 ± 0.07 d	7.93 ± 0.08 a	8.70 ± 0.09 b	<0.001	<0.001	>0.050
PCP	11.06 ± 0.15 e	10.16 ± 0.11 c	10.17 ± 0.08 c	10.50 ± 0.08 d	8.78 ± 0.10 a	9.80 ± 0.10 b	<0.001	<0.001	>0.050
BW	61.98 ± 2.23 cd	64.21 ± 1.51 d	59.43 ± 1.77 c	73.35 ± 1.23 e	42.29 ± 1.18 a	52.74 ± 1.58 b	<0.001	<0.001	0.001

a–f.—Different superscripts in the same row indicate significant differences between genetic lines. Head length (HL), Head width (HW), Wool extension on forehead “bow” (WF), Wool extension on cheeks (WCh), Ear length (EL), Shoulder width (SW), Neck length (NL), Lumbar back line length (LBL), Rump width (RW), Rump length (RL), Buttock width (BuW), Buttock length (BL), Wither height (WH), Bicostal diameter (BD), Height at substernal foramen (HSF), Dorso-sternal diameter (DSL), Longitudinal diameter (LD), Rump height (RH), Thoracic perimeter (TP), Anterior and posterior cannon bone perimeter (ACP and PCP), and Body weight (BW).

**Table 5 animals-13-00313-t005:** Means and standard errors for zoometric indexes taken for animals belonging to the different historical genetic lines of Merino Sheep breed. Analysis of variance (*p*-value): gender and genetic line effects.

	Genetic Lines						*p*-Value		
	Donoso	Egea	Granda	Hidalgo	Maesso	López- Montenegro	Line	Gender	Line × Gender
BI	72.20 ± 0.77 a	78.41 ± 0.75 b	81.02 ± 0.47 cd	82.77 ± 0.45 d	79.78 ± 0.44 bc	81.67 ± 0.49 cd	<0.001	<0.001	<0.001
CoI	85.54 ± 2.51 c	90.29 ± 1.59 d	84.81 ± 1.95 c	97.73 ± 1.26 e	68.23 ± 1.23 a	73.83 ± 1.67 b	<0.001	<0.001	<0.001
PeI	83.06 ± 0.86 c	78.06 ± 0.51 ab	78.24 ± 0.49 ab	76.95 ± 0.56 a	79.41 ± 0.49 b	77.77 ± 0.59 ab	<0.001	<0.001	<0.001
ThI	102.90 ± 0.18 a	103.78 ± 0.50 ab	105.07 ± 0.38 bc	106.22 ± 0.37 c	102.75 ± 0.37 a	105.76 ± 0.34 c	<0.001	<0.001	>0.050
CeI	42.98 ± 0.30 a	45.94 ± 0.25 b	45.93 ± 0.16 b	43.65 ± 0.15 a	45.95 ± 0.24 b	45.69 ± 0.20 b	<0.001	<0.001	<0.001
PI	100.71 ± 0.86 de	98.69 ± 0.50 cd	96.28 ± 0.62 ab	94.99 ± 0.47 a	97.53 ± 0.50 bc	101.62 ± 0.56 e	<0.001	0.005	<0.001
LCI	15.79 ± 0.38 c	14.32 ± 0.20 b	15.91 ± 0.36 c	13.01 ± 0.14 a	19.29 ± 0.36 e	17.15 ± 0.36 d	<0.001	<0.001	<0.001

a–e.—Different superscripts in the same row indicate significant differences between genetic lines. Body Index (BI) Compactness Index (CoI), Pelvic Index (PeI), Thoracic Index (ThI), Cephalic Index (CeI), Proportionality Index (PI) and Load Cannon Bone Index (LCI).

**Table 6 animals-13-00313-t006:** Principal components, total and accumulated variance, and factor and factor loadings for the morphometric measurements in the six genetic lines.

	Components		
Traits and Indexes	Factor 1	Factor 2	Communality
HL	0.930	−0.089	0.873
HW	0.855	−0.115	0.745
WF	0.820	−0.239	0.730
WCh	0.359	−0.377	0.272
EL	0.485	0.788	0.856
SW	0.942	−0.142	0.908
NL	0.810	0.476	0.883
WH	0.936	0.093	0.885
BD	0.953	−0.011	0.909
HSF	0.804	0.263	0.716
DSD	0.934	−0.055	0.875
LD	0.930	0.142	0.885
LBL	0.720	0.304	0.611
RH	0.942	0.114	0.901
RW	0.860	−0.023	0.741
RL	0.965	−0.005	0.931
BuW	0.805	−0.266	0.719
BL	0.897	0.066	0.808
ThP	0.910	−0.170	0.857
ACP	0.806	−0.246	0.881
PCP	0.888	−0.287	0.870
BW	0.961	−0.004	0.923
Total variance (%)	73.80	7.01	
Cumulative variance (%)	73.80	80.81	

**Table 7 animals-13-00313-t007:** Discriminant analysis: percentage of animals correctly classified in their historical genetic lines.

Predicted/Actual Membership	Hidalgo	Maesso	Egea	López-Montenegro	Granda	Donoso
Hidalgo	95.0	0	1.3	0	3.8	0
Maesso	0	96.7	0	3.3	0	0
Egea	0	0	70.7	4.9	24.4	0
López-Montenegro	0	1.5	12.1	80.3	6.1	0
Granda	5.0	0	18.3	15.0	61.7	0
Donoso	0	0	0	0	0	100

## Data Availability

Not applicable.

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
