# Peer review of "Morphostructural Differences between the Historical Genetic Lines of the Spanish Merino Sheep"

_animals, 2023, doi:10.3390/ani13020313_

Round 1

Reviewer 1 Report

In the introduction  , it should be supplemented to use genome data to reveal the genetic differentiation results of Spanish Merino sheep

Author Response

In the introduction, it should be supplemented to use genome data to reveal the genetic differentiation results of Spanish Merino sheep.

Thank you for your comments. As the referee says, our study should be supported with the genomic results. We referenced the genomic differences between the lines (lines 79-80) supported by a previous work: Granero, A., et al., Genomic Population Structure of the Main Historical Genetic Lines of Spanish Merino Sheep. Animals, 2022. 12(10): p. 1327. To clarify this further, we have extended the text in the introduction (lines 82-86):

“The genomic results also revealed a different selection footprint compared with the modern Merino, together with the existence of more ancestral selection processes in these genetic lines compared to modern Merino sheep, demonstrating that many generations were selected earlier to avoided the current practice of crossbreeding [12].”

We also widened the discussion along the following lines (lines 398-400):

“This result reinforces that obtained by Granero [12], in which the Donoso was the line most genetically differentiated from the others according to the principal component analysis and the genetic structure of the population.”

The phrase in line 402-405 did not have its corresponding reference – this has now been added:

“This could be explained by the fact that all these lines are currently bred in the same geographical area and under similar management. Therefore, it can be assumed that these lineages could have been formed centuries ago from the same historical herds [12]”

…and in line 415-418

“In this case, the morphometric approach in the results is different from the genomic approach. According to the genetic differences, the López-Montenegro and Granda lines show the highest similarity levels, and Donoso is the most differentiated line, rather than the Maesso line [12]”

Reviewer 2 Report

Although "old-fashioned," it is an interesting study, well done from the standpoint of results and statistical analysis. This study has a practical spin-off in good breeding standards.
However, I suggest a small grammar check and the more orderly arrangement of the tables.

Author Response

Although "old-fashioned," it is an interesting study, well done from the standpoint of results and statistical analysis. This study has a practical spin-off in good breeding standards.

However, I suggest a small grammar check and the more orderly arrangement of the tables.

Thank you very much for your comments. The English has now been checked and the tables have been arranged better.

Reviewer 3 Report

The bolding of text is not necessary.

In a study of phenotypic differentiation, it is important to recognize that phenotypes result from the sum of genotype and environment. Is there some confounding of flocks with the strains? If so, the phenotypic differences might be due to the environment of the flock rather than the strain.

The prevalence of significant line x gender interaction effects on the phenotypes merits additional attention. One would hope the multivariate results were consistent between males and females. However, this hypothesis must be tested. 

Including the ratio indexes in the PCA implies a degree of non-linearity of effects because the linear measures that are numerator and denominator are also included separately. This should be explained. In addition to the first two eigenvectors, it would be useful to know the eigenvalues for the principal components. Explain why only the first two principal components are presented. 

Some references are made to "cattle" in discussing the results (lines 298, 350 and 351). These would seem to be unintended.

Table 4: do not use "," for decimal point.

Some of the results cast doubt upon the "purity" of some strains. In the context of genetic conservation this merits addressing. Also, a direct comparison of these results with the previously published genomic analysis should be made.  

Author Response

The bolding of text is not necessary.

Many thanks – we had not noticed this mistake before the manuscript was uploaded. The bold text in lines 53, 54, 79, 80 and 82 has now been taken away.

In a study of phenotypic differentiation, it is important to recognize that phenotypes result from the sum of genotype and environment. Is there some confounding of flocks with the strains? If so, the phenotypic differences might be due to the environment of the flock rather than the strain.

Thank you for your comment. We agree with the reviewer about the importance of the ambience in the phenotype of the animals, but the remaining animals of these ancestral lines are grouped on only 11 farms, all of which have been used in the study. We have added a phrase to clarify this (lines 61-62):

 “These lines are the only remaining ancestral sheep from the original Merino and are currently bred in just 11 flocks.”

The prevalence of significant line x gender interaction effects on the phenotypes merits additional attention. One would hope the multivariate results were consistent between males and females. However, this hypothesis must be tested. 

We greatly appreciate the reviewer's comment, and we agree. However, the correct interpretation of a significant interaction requires linear comparisons of one degree of freedom. Only in this way is it possible to isolate and exhaust the meaning of the interaction. However, this was not the aim of this study, and for this reason we have not gone into it in any depth.

Including the ratio indexes in the PCA implies a degree of non-linearity of effects because the linear measures that are numerator and denominator are also included separately. This should be explained. In addition to the first two eigenvectors, it would be useful to know the eigenvalues for the principal components. Explain why only the first two principal components are presented.

As recommended, we have excluded the morphometry indices from the Principal Components Analysis, and the new information is presented in the study. We have used only two PCs because they cover the most variables and are used in this way in most studies. If in the extraction method we use the premise of an Eigenvalue greater than 1, this produces three main components, of which the third component only explains 4.56% of the variance.

Some references are made to "cattle" in discussing the results (lines 298, 350 and 351). These would seem to be unintended.

Thank you for noticing this. It was, of course, a mistake. In lines 298, 350 and 351, “cattle” has been changed to “sheep”.

Table 4: do not use "," for decimal point.

Thank you for this comment. We have corrected this in Table 4.

Some of the results cast doubt upon the "purity" of some strains. In the context of genetic conservation this merits addressing. Also, a direct comparison of these results with the previously published genomic analysis should be made.  

Thank you for this comment. We have now added a sentence referring to the genomic approach (see lines 405-408) and we have attempted to clarify why the Granda, Egea and López-Montenegro appear not to be “pure” lines with the phrase:

“This could be explained by the fact that all these lines are currently bred in the same geographical area and under similar management. It can therefore be assumed that these lineages could have been formed centuries ago from the same historical herds [12]. Also, the management used by the breeders of these lines has always aimed to conserve the variability within each of their closed herds, avoiding the high-pressure levels of selection so the animals could have more morphometric similarities within the herd although they are genetically different.”

Reviewer 4 Report

This is an interesting study that is important for organizing the effective conservation of the Merino breed in Spain, which is the most important genetic reserve of the breed. One additional analysis would be to separate out the different farms to see if farm-of-origin provided any differences along with line of breeding. This may explain some of the divergence within the Maesso line, and this could affect conservation recommendations by splitting the line. This is especially important to consider for the Maesso sheep that cluster away from all others.

Author Response

This is an interesting study that is important for organizing the effective conservation of the Merino breed in Spain, which is the most important genetic reserve of the breed. One additional analysis would be to separate out the different farms to see if farm-of-origin provided any differences along with line of breeding. This may explain some of the divergence within the Maesso line, and this could affect conservation recommendations by splitting the line. This is especially important to consider for the Maesso sheep that cluster away from all others.

Thank you for your comment. We agree with the reviewer about the importance of the ambience in the phenotype of the animals, although the remaining animals from these ancestral lines are now found on only 11 farms, all of which were used in the present study. This is the reason why we cannot compare the lines in different flocks. We have added a phrase to clarify this (lines 61-62):

 “These lines are the only remaining ancestral sheep from the original Merino and are currently bred in just 11 flocks.”

Round 2

Reviewer 3 Report

The concerns raise in the previous review are still of concern in the revised manuscript.